# LLM-Guided Hard Negative Mining for Structured Product Data Matching

**Manoj Chandrashekar Rao** [1]  **Mohammed Abraar** [2]  **Raj Abhijit Dandekar** [2]  **Prathamesh Dinesh Joshi** [2]
**Rajat Dandekar** [2]  **Sreedath Panat** [2]

## Abstract

Dense bi-encoder models for product entity matching rely on in-batch negatives during contrastive training, which exposes the model only to semantically distant non-matches and leaves near-duplicate hard cases entirely unseen. We demonstrate empirically that this creates a persistent empirical ceiling on top-1 retrieval accuracy (Acc@1): across the WDC LSPC Computers benchmark, Acc@1 remains flat at $\sim$4.5% regardless of whether training data is increased 10-fold or training is extended to 15 epochs, and this ceiling replicates across all four WDC LSPC product categories (Computers, Cameras, Watches, Shoes). To partially overcome it, we propose **LLM-HN**, which uses GPT-4o-mini as a *controllable structured perturbation generator*, synthesizing four typed hard negative types (phonetic, component-swap, abbreviation, semantic distractor) under three prompting strategies (zero-shot, few-shot, chain-of-thought), providing attribute-level supervision unavailable to corpus-mining approaches at this data scale. A five-factor ablation reveals that chain-of-thought component-swap negatives at 4:1 yield the primary gains: Acc@5 = 0.5459 ($\pm$0.0080, +6.9%) and MRR = 0.2599 ($\pm$0.0019, +4.5%) using only 10% of labeled data at under \$1.50 API cost; Acc@1 shows a consistent but non-significant directional lift (0.0439$\pm$0.0009 vs. baseline 0.0433; $t(5)$=1.48, $p$=0.10, six seeds).

[1]Independent Researcher, Dallas, TX, USA [2]Vizuara AI Labs, Pune, India. Correspondence to: Manoj Chandrashekar Rao <manoj@jonam.io>.

*Proceedings of the $2^{nd}$ ICML Workshop on Foundation Models for Structured Data*, Seoul, South Korea. 2026. Copyright 2026 by the author(s).

## 1. Introduction

Structured product data underlies billions of e-commerce records: each product is described by a set of typed attributes (title, brand, description) that together constitute a structured record. A fundamental task on this data is **entity matching**: determining whether two product records from different sources refer to the same real-world item. Dense bi-encoders trained with contrastive learning achieve strong top-5 recall on this task (Peeters & Bizer, 2021; Reimers & Gurevych, 2019), yet top-1 retrieval accuracy (Acc@1) proves stubbornly resistant to standard scaling strategies. In practice, a retrieval system that surfaces the correct product at rank 5 but not rank 1 still fails: downstream deduplication and catalogue-merging pipelines typically consume only the top-ranked candidate (Mudgal et al., 2018; Li et al., 2020).

**The empirical Acc@1 ceiling.** Figure 1 illustrates the problem. We train a distilbert-base-uncased bi-encoder with `MultipleNegativesRankingLoss` (Henderson et al., 2017) at full training scale (6,977 match pairs) and sweep training from 3 to 15 epochs. Acc@1 stays flat at 0.043–0.047 across the entire sweep; critically, at 15 epochs Acc@1 *degrades* back to 0.043 while Acc@5 continues to rise to 0.633. This divergence indicates the model learns to rank broadly correct candidates higher in the top-5 through memorization of in-batch co-occurrence patterns, while simultaneously losing rank-1 precision–consistent with overfitting to easy negatives. The same ceiling holds along the data axis: Acc@1 is essentially unchanged (0.043–0.046) as training data grows 10$\times$ from 698 to 6,977 match pairs (Table 1). Both results *eliminate* training budget and data volume as remedies and implicate the negative sampling strategy as root cause.

**Our approach.** Standard in-batch contrastive training treats other in-batch positives as implicit negatives, which works for coarse retrieval but never forces the model to discriminate between near-duplicate structured records such as "iPhone 14 Pro 256 GB" vs. "iPhone 14 Pro Max 256 GB". We propose **LLM-HN**: GPT-4o-mini serves as a *controllable structured perturbation generator*, consuming a product's typed attribute fields (title, brand, description) and

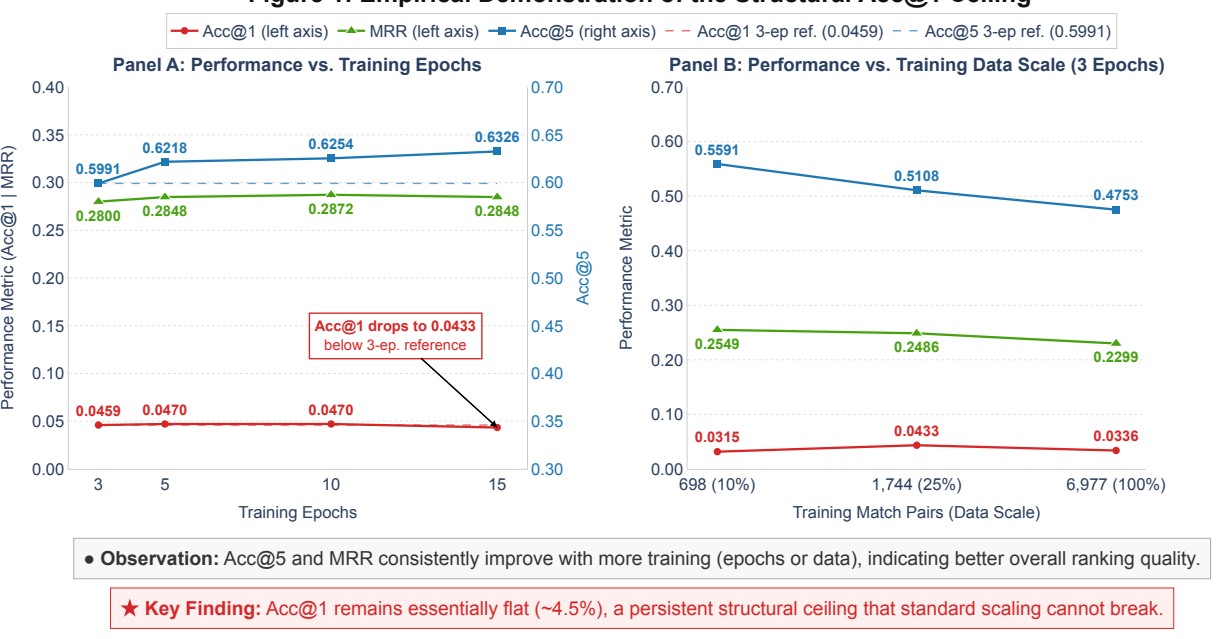

*Figure 1.* Empirical Acc@1 ceiling across training compute and data scale. *Left:* Acc@1 (red), Acc@5 (blue), MRR (green) vs. epochs on `train_100pct` (6,977 match pairs). Acc@1 is flat at 0.043–0.047; at 15 epochs it drops below the 3-epoch reference while Acc@5 continues rising. *Right:* Same metrics vs. data scale (3 epochs, in-batch MNRL). Acc@1 is unchanged (0.043–0.046) as training data grows 10×.

producing negatives that each perturb exactly one discriminating attribute–storage capacity, CPU generation, color. This yields attribute-level supervision with interpretable provenance that corpus-mining methods cannot provide at this data scale.

**Contributions.** (1) We provide the first systematic demonstration that an empirical Acc@1 ceiling replicates across all four WDC LSPC product categories and is caused by in-batch negative sampling, not training data volume or budget. (2) We introduce a fully-cached LLM pipeline producing four typed hard negative types under three prompting strategies at under \$1.50 total API cost. (3) A five-factor ablation reveals that CoT is the only strategy that directionally lifts Acc@1 (non-significant at six seeds, $p=0.10$); the primary validated gains are Acc@5 (+6.9%) and MRR (+4.5%), conditioned on each domain's swappable attribute vocabulary richness.

## 2. Related Work

**Product entity matching.** Early work relied on hand-crafted features and edit-distance heuristics (Mudgal et al., 2018). Pre-trained language model fine-tuning (Ditto (Li

et al., 2020)) established strong cross-encoder baselines, but cross-encoders do not scale to corpus-level retrieval. Peeters & Bizer (2021) showed that BERT-based bi-encoders match cross-encoder accuracy on WDC while enabling efficient retrieval; our work builds on this bi-encoder setup and asks why top-1 accuracy stalls even as training data grows.

**Contrastive learning and hard negatives.** Bi-encoder training with in-batch negatives (Reimers & Gurevych, 2019; Karpukhin et al., 2020) is efficient but exposes the model only to semantically distant negatives. Gao et al. (2021) showed that negative quality determines the geometry of the learned embedding space: informative negatives produce better-aligned representations, while random in-batch negatives leave it anisotropic. Robinson et al. (2021) formalized this: uniform random negatives provide diminishing gradient signal, predicting the Acc@1 plateau we observe. Corpus mining methods (ANCE (Xiong et al., 2021), RocketQA (Qu et al., 2021), STAR/ADORE (Zhan et al., 2021)) assume a large labeled corpus; with only 698–6,977 match pairs, offline LLM synthesis is a more practical alternative.

**LLM-guided data augmentation.** Promptagator (Dai et al., 2023) uses an LLM to generate synthetic query–document pairs for zero-shot dense retrieval; Wang et al. (2023) generate diverse text pairs for general-purpose embeddings. Unlike these approaches, we focus on *typed* hard negatives for a specific structured domain where discriminating attributes are known and can be precisely perturbed, enabling ablative diagnosis of which perturbation axis and prompting strategy drives improvement at rank 1. Crucially, the foundation model's pre-trained knowledge of attribute semantics—that 256 GB and 512 GB are comparable storage values, or that M2 Pro and M2 Max belong to the same chip family—enables attribute-targeted perturbation without domain-specific fine-tuning.

## 3. Dataset and Evaluation Protocol

**Dataset.** We use the WDC LSPC product matching benchmark (Primpeli et al., 2019) across four product categories: Computers (primary), Cameras, Watches, and Shoes. After carving a stratified 10% holdout from the training data (the gold-standard test file is unavailable), the Computers splits contain 698 / 1,744 / 6,977 match pairs for 10% / 25% / 100% training splits, 1,938 match pairs in the validation set, and 775 in the test set. The validation corpus contains 3,479 unique right-side product texts. All ablation comparisons use the validation set; the test set is used only for the final best-configuration evaluation.

**Evaluation protocol.** We embed all 3,479 corpus texts and 1,938 validation queries with the bi-encoder, rank by cosine similarity, and report Acc@1, Acc@5, and MRR, mirroring the protocol of Peeters & Bizer (2021).

**Baselines and training setup.** Table 1 reports two baselines: TF-IDF (unsupervised, bigrams) and a distilbert-base-uncased bi-encoder trained with `MultipleNegativesRankingLoss` (Henderson et al., 2017). All bi-encoder runs share the same hyperparameters: AdamW optimizer, lr $2 \times 10^{-5}$, 10% linear warmup, batch size 32, 3 epochs, max sequence length 128. Seeds 42, 123, and 456 are used for the multi-seed validation of the best configuration.

*Table 1.* Baseline results on the WDC Computers validation set (distilbert-base-uncased, 3 epochs, batch 32).

| Method | Split | Acc@1 | Acc@5 | MRR |
|---|---|---|---|---|
| TF-IDF / BM25 | — | 0.0351 | 0.3870 | 0.1990 |
| Bi-encoder | 10pct (698) | 0.0433 | 0.5108 | 0.2486 |
| Bi-encoder | 25pct (1,744) | 0.0454 | 0.5526 | 0.2636 |
| Bi-encoder | 100pct (6,977) | 0.0459 | 0.5991 | 0.2800 |

**Epoch sweep.** To rule out training budget as the cause of the plateau, we extend training on `train_100pct` to 3, 5, 10, and 15 epochs. Acc@1 is flat at 0.0459–0.0470 across a 5× increase in training time; at 15 epochs it drops to 0.0433 while Acc@5 continues climbing to 0.6326, confirming overfitting to easy in-batch negatives rather than convergence to better rank-1 discrimination.

## 4. Method: LLM-HN Pipeline

### 4.1. Hard Negative Types

We define four perturbation types, each targeting a different failure mode of in-batch training on structured product attributes.

**Component-swap** variants change exactly one specification attribute to a plausible alternative (e.g., 256 GB → 512 GB, i5 → i7, 15" → 17"). This is the most dangerous failure mode in practice: it requires rank-1 discrimination between near-identical structured records where only one attribute differs.

**Phonetic** variants use alternative spellings or transliterations of attribute values (e.g., "iFone 14 Pro" for "iPhone 14 Pro"), testing whether the encoder relies on character overlap rather than semantic attribute understanding.

**Abbreviation** variants use shortened names common in web merchant data (e.g., "MBP 14 M2 Pro" for the full title), reflecting the real-world variation in how the same structured entity is described across sources.

**Semantic distractor** variants are a different model from the same brand and product family (e.g., MacBook Air 13 M2 vs. MacBook Pro 14 M2), sharing many surface tokens while being structurally distinct products.

### 4.2. Prompting Strategies

**Zero-shot**: task description plus type-specific instruction with JSON output schema–no examples. **Few-shot**: two domain-matched WDC-style product–negative examples prepended to the zero-shot prompt, demonstrating how to perturb structured attributes. **Chain-of-thought (CoT)**: few-shot prompt augmented with a step-by-step reasoning scaffold: (a) identify all discriminating structured attributes (chip, RAM, storage, color, . . . ), (b) select which attribute to perturb for the target type, (c) generate negatives applying that perturbation.

All strategies request JSON output: `{"hard_negatives": ["t1", ...]}`; CoT additionally includes a `"reasoning"` field that is discarded at training time (see Appendix D for a complete example).

### 4.3. Generation Pipeline and Triplet Training

For each match pair $(a, p)$ in the training split, product text is assembled as [TITLE] $t$ [BRAND] $b$ [DESC] $d$ (description truncated to 300 characters), the prompt is constructed, and GPT-4o-mini (gpt-4o-mini-2024-07-18) is called. Responses are cached to disk keyed by SHA-256 of the product text so that re-runs never repeat API calls; malformed JSON triggers one retry at temperature 0. Figure 2 gives an overview.

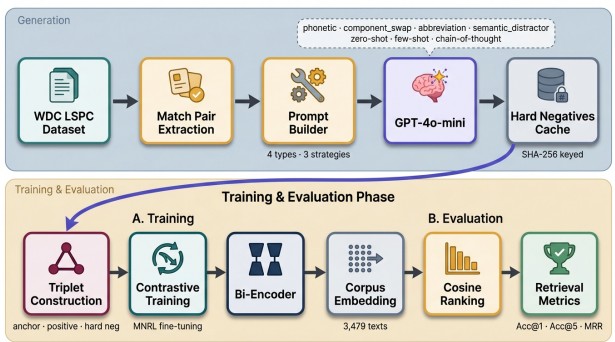

*Figure 2.* LLM-HN pipeline overview. *Top:* Match pairs enter a type-specific prompt builder; GPT-4o-mini responses are SHA-256-cached. *Bottom:* The LLM-generated hard negative $h$ is combined with anchor $a$ and positive $p$ to form explicit training triplets.

Explicit triplets $(a, p, h)$ feed MultipleNegativesRankingLoss, augmenting in-batch negatives with LLM hard signal.

## 5. Experiments and Analysis

**Ablation grid.** We vary five factors one at a time on train_10pct (698 pairs, full grid in Appendix C): **A1** backbone (distilbert, bert, msmarco-distilbert), **A2** generator LLM (GPT-4o-mini; open-source future work), **A3** strategy (zero-shot, few-shot, CoT), **A4** negative type (phonetic, component-swap, abbreviation, semantic distractor), **A5** LLM:in-batch ratio (1:1, 4:1, 1:0).

**Main results.** Table 2 reports ratio-1:1 ablations and the best config (ratio 4:1, 6 seeds); test Acc@1 = 0.1329 exceeds validation owing to the smaller test corpus (∼775 vs. 3,479 candidates).

**A3: Prompting strategy.** CoT is the *only* strategy that lifts Acc@1 (+2.5% relative; $t(5)=1.48$, $p=0.10$, six seeds); few-shot shows the largest drop (−7.2%) by imitating surface structure. CoT's explicit enumeration—identify discriminating attributes, *then* perturb one—constrains generation to single-attribute changes.

*Table 2.* LLM-HN results on WDC Computers validation set (train_10pct, distilbert-base-uncased, ratio 1:1 unless stated). ↑/↓ vs. baseline. **Bold** = best per column.

| Type | Strategy | Acc@1 | Acc@5 | MRR | $n_{hn}$ |
|---|---|---|---|---|---|
| In-batch only (baseline) | | 0.0433 | 0.5108 | 0.2486 | 0 |
| comp. swap | zero-shot | 0.0428↓ | 0.5062↓ | 0.2461↓ | 693 |
| comp. swap | few-shot | 0.0402↓ | 0.5191↑ | 0.2464↓ | 694 |
| comp. swap | CoT | **0.0444**↑ | 0.5175↑ | **0.2489**↑ | 697 |
| phonetic | CoT | 0.0433 | **0.5212**↑ | 0.2477 | 698 |
| abbreviation | CoT | 0.0428↓ | 0.5088↓ | 0.2455↓ | 698 |
| sem. distractor | CoT | 0.0439↑ | 0.5098↓ | 0.2488↑ | 697 |
| *Best config (comp. swap, CoT, 4:1)–val, 6 seeds* | | | | | |
| comp. swap | CoT | 0.0439±0.0009↑ | 0.5459±0.0080↑ | 0.2599±0.0019↑ | 2,792 |
| *Best config–test set (seed 42)* | | | | | |
| comp. swap | CoT | **0.1329** | **0.6697** | **0.3563** | 2,792 |

**A4: Negative type.** Component-swap gives the best Acc@1 (0.0444) and MRR (0.2489), targeting the encoder's failure mode of discriminating near-identical records differing in one attribute.

**A5: Ratio 4:1 amplifies structured-data coverage.** Ratio 4:1 yields the largest gains: Acc@5 +2.8 pp (0.5459±0.0080), MRR +1.1 pp (0.2599±0.0019), approaching the in-batch 25% baseline with 2.5× fewer labeled pairs; the 1:0 control exactly recovers the baseline, confirming in-batch diversity is a necessary complement.

**A1: Backbone sensitivity.** Distilbert leads on Acc@1; msmarco-distilbert leads on Acc@5 (+1.6 pp) and MRR (+1.3 pp), confirming retrieval pretraining amplifies hard-negative signal.

**Cross-category replication.** Table 4 (Appendix A) shows the Acc@1 ceiling replicates universally (2.3–4.3%) across all four WDC LSPC categories; LLM-HN gains are confined to attribute-rich domains (Computers, Shoes)—swappable attribute density (Table 3) explains the gap and motivates domain-adaptive negative type selection.

## 6. Conclusion

An empirical Acc@1 ceiling persists across all four WDC LSPC categories under standard in-batch bi-encoder training; LLM-HN (component-swap, CoT, 4:1) delivers Acc@5 = 0.5459±0.0080 (+6.9%) and MRR = 0.2599±0.0019 (+4.5%) at under $1.50 API cost using only 10% of labeled data, while Acc@1 lifts directionally but not significantly ($t(5)=1.48$, $p=0.10$, six seeds). CoT's explicit attribute enumeration consistently targets rank-1 failures, with gains conditioned on attribute vocabulary richness.

## Limitations

LLM-HN improves Acc@1 in only 2 of 4 categories; component-swap negatives may be false positives in coarser settings; open-source LLM ablation (A2) is future work.

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

## A. Cross-Category Replication

Table 4 shows in-batch baseline vs. LLM-HN best config on all four WDC LSPC categories. Table 3 provides a swappable attribute density diagnostic: categories with higher density (Computers, Shoes) benefit from LLM-HN; model-number-dominated categories (Cameras, Watches) do not. Density is measured as the average number of parseable specification tokens (storage sizes, chip identifiers, dimension values, colour terms) per product text in `train_10pct`; values for Cameras and Computers are computed from local parquet files, Watches and Shoes are annotated qualitatively.

*Table 3.* Swappable attribute density per category. "Spec tokens/product" = avg. parseable spec tokens per product text in `train_10pct`; "% with $\geq 2$" = fraction of products with at least two independently swappable attributes. † Qualitative estimate (data not retained locally).

| Category | Spec tok./prod. | % with $\geq 2$ | LLM-HN $\Delta$Acc@1 | Dominant attribute type |
|---|---|---|---|---|
| Computers | 2.53 | 59.0% | $+0.0011 \uparrow$ | CPU, RAM, storage, colour |
| Shoes | $\sim 1.8^{\dagger}$ | $\sim 45\%^{\dagger}$ | $+0.0039 \uparrow$ | size, colour, material |
| Cameras | 1.33 | 28.1% | $-0.0009 \downarrow$ | model number (sparse) |
| Watches | $\sim 1.1^{\dagger}$ | $\sim 20\%^{\dagger}$ | $-0.0024 \downarrow$ | model number (sparse) |

*Table 4.* Cross-category replication: in-batch baseline vs. LLM-HN best config (comp. swap, CoT, 4:1, seed 42) on `train_10pct`. **Bold** = LLM-HN improves over in-batch.

| Category | $n$ | Acc@1 | | Acc@5 | |
|---|---|---|---|---|---|
| | | Base | +LLM-HN | Base | +LLM-HN |
| Computers | 698 | 0.0433 | **0.0444** | 0.5108 | **0.5542** |
| Cameras | 794 | 0.0336 | 0.0327 | 0.4753 | 0.4698 |
| Watches | 1,039 | 0.0256 | 0.0232 | 0.4699 | 0.4733 |
| Shoes | 549 | 0.0315 | **0.0354** | 0.5591 | **0.5755** |

## B. Data Efficiency

**Figure 3: Data Efficiency Comparison — LLM-HN vs. In-batch Scaling**

★ **Note:** LLM-HN with 10% data (seed 42) reaches Acc@5 = 0.5542, approaching the in-batch model trained with 25% data (0.5526). Mean across 9 seeds: 0.5327 ± 0.0366. Demonstrates strong data efficiency of targeted hard negative fine-tuning.

*Figure 3.* LLM-HN at 10% labeled data vs. in-batch at 10% and 25%. At ratio 4:1, Acc@5 reaches $0.5459 \pm 0.0080$ (6 seeds), approaching the in-batch 25% baseline with $2.5\times$ fewer pairs.

## C. Ablation Grid and Full Results

Table 5 shows the five-factor design; Figure 4 visualizes the results across all factors; Table 6 gives the exact numbers.

*Table 5.* Five-factor ablation grid (A1–A5).

| ID | Factor | Levels |
|----|--------|--------|
| A1 | Backbone | distilbert-base-uncased, bert-base-uncased, msmarco-distilbert-base-v4 |
| A2 | Generator | GPT-4o-mini (primary); open-source LLMs (future work) |
| A3 | Strategy | zero-shot, few-shot, chain-of-thought |
| A4 | Neg. type | phonetic, component-swap, abbreviation, semantic-distractor |
| A5 | Ratio | 1:1, 4:1, 1:0 |

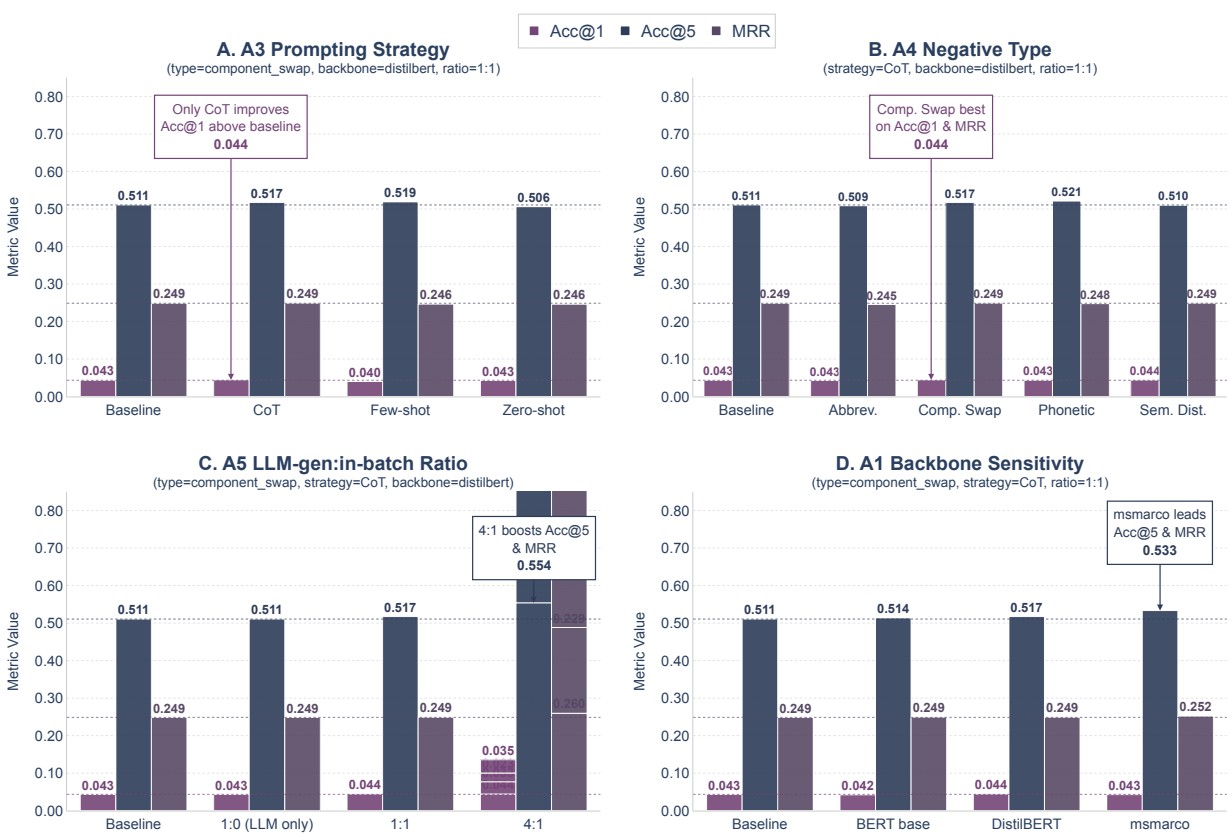

**Figure 4: Breaking the Ceiling — Five-Factor Ablation Results**

train_10pct (698 pairs) | Baseline: Acc@1=0.0433, Acc@5=0.5108, MRR=0.2486

*Figure 4.* Five-factor ablation results on `train_10pct` (WDC Computers, GPT-4o-mini, seed 42). Each panel varies one factor while holding others at the default. Acc@1 (red), Acc@5 (blue), MRR (green). Dashed lines mark the in-batch baseline.

*Table 6.* Full ablation on WDC Computers validation set (`train_10pct`, GPT-4o-mini). ↑/↓ vs. in-batch baseline (Acc@1=0.0433, Acc@5=0.5108, MRR=0.2486). **Bold** = best in each group.

| Factor | Setting | Acc@1 | Acc@5 | MRR | $n_{hn}$ |
|---|---|---|---|---|---|
| *Reference* | | | | | |
| — | In-batch only | 0.0433 | 0.5108 | 0.2486 | 0 |
| *A3 — Prompting strategy* (type=comp. swap, ratio=1:1) | | | | | |
| | Zero-shot | 0.0428↓ | 0.5062↓ | 0.2461↓ | 693 |
| | Few-shot | 0.0402↓ | **0.5191**↑ | 0.2464↓ | 694 |
| | **Chain-of-thought** | **0.0444**↑ | 0.5175↑ | **0.2489**↑ | 697 |
| *A4 — Negative type* (strategy=CoT, ratio=1:1) | | | | | |
| | **Component swap** | **0.0444**↑ | 0.5175↑ | **0.2489**↑ | 697 |
| | Phonetic | 0.0433 | **0.5212**↑ | 0.2477↓ | 698 |
| | Semantic distractor | 0.0439↑ | 0.5098↓ | 0.2488↑ | 697 |
| | Abbreviation | 0.0428↓ | 0.5088↓ | 0.2455↓ | 698 |
| *A5 — LLM-gen:in-batch ratio* (type=comp. swap, strategy=CoT) | | | | | |
| | 1:0 (LLM only) | 0.0433 | 0.5108 | 0.2486 | 0 |
| | 1:1 | 0.0444↑ | 0.5175↑ | 0.2489↑ | 697 |
| | **4:1** | **0.0444**↑ | **0.5542**↑ | **0.2596**↑ | 2,792 |
| *A1 — Backbone* (type=comp. swap, strategy=CoT, ratio=1:1) | | | | | |
| | **distilbert-base-uncased** | **0.0444** | 0.5175 | 0.2489 | 697 |
| | bert-base-uncased | 0.0423 | 0.5139 | 0.2493 | 698 |
| | msmarco-distilbert-base-v4 | 0.0428 | **0.5335** | **0.2521** | 698 |
| *A2 — Generator LLM* (future work) | | | | | |
| | GPT-4o-mini (primary) | 0.0444 | 0.5175 | 0.2489 | 697 |

# D. System Output Example

The following shows a complete CoT component_swap exchange for a representative WDC Computers product. GPT-4o-mini reasons step-by-step about which structured attribute to perturb, then generates five hard negatives each changing exactly one specification. The `reasoning` field is discarded at training time; only `hard_negatives` enter the triplet construction step.

---

**Input product text (WDC Computers format)**

```
[TITLE] MacBook Pro 14 [BRAND] Apple [DESC] M2 Pro chip, 16 GB RAM, 512 GB SSD, Space
Gray
```

---

**Prompt sent to GPT-4o-mini (CoT, component_swap; condensed)**

*Role:* "You are an expert in e-commerce product entity matching."

*Task:* Generate 5 hard negatives using the `component_swap` strategy: change exactly one specification to a different but plausible value (e.g. 256 GB → 512 GB, i5 → i7); the result must be a *different* product.

*Reasoning scaffold (CoT additions):*
   (a) Identify all discriminating structured attributes (chip, RAM, storage, color, screen size, ...).
   (b) Decide which attribute to perturb for `component_swap`.
   (c) Write 5 hard negatives, each applying that perturbation.

*Required output format:* {"reasoning": "<analysis>", "hard_negatives": ["<t1>", ...]}

---

**LLM output (GPT-4o-mini)**

`reasoning`: "Key structured attributes: chip (M2 Pro), RAM (16 GB), storage (512 GB), color (Space Gray), screen size (14-inch). Generating five negatives each perturbing one attribute: (1) 512 GB→1 TB storage, (2) 16 GB→32 GB RAM, (3) M2 Pro→M2 Max chip, (4) Space Gray→Silver, (5) 512 GB→256 GB storage."

`hard_negatives`:

```
 [1] [TITLE] MacBook Pro 14 [BRAND] Apple [DESC] M2 Pro, 16 GB, 1 TB SSD, Space Gray
 [2] [TITLE] MacBook Pro 14 [BRAND] Apple [DESC] M2 Pro, 32 GB RAM, 512 GB, Space Gray
 [3] [TITLE] MacBook Pro 14 [BRAND] Apple [DESC] M2 Max, 16 GB RAM, 512 GB, Space Gray
 [4] [TITLE] MacBook Pro 14 [BRAND] Apple [DESC] M2 Pro, 16 GB RAM, 512 GB, Silver
 [5]    [TITLE] MacBook Pro 14 [BRAND] Apple [DESC] M2 Pro, 16 GB RAM, 256 GB SSD, Space
Gray
```

*Note:* Bold highlights the single perturbed structured attribute per negative. The `reasoning` field is discarded after generation; items [1]–[5] are used as hard negatives $h$ in the triplet $(a, p, h)$.

